# Non-Normality as a Predictor of Participation in Bullying: Valuation in Victims and Aggressors

**DOI:** 10.3390/ijerph19106344

**Published:** 2022-05-23

**Authors:** Raúl Carretero Bermejo, Alberto Nolasco Hernández, Laura Gracia Sánchez

**Affiliations:** 1Psychology Department, Education Faculty of Ciudad Real, University of Castilla la Mancha, Ronda de Calatrava, n° 3, 13071 Ciudad Real, Spain; 2Pedagogy Department, Human Sciences Faculty of Teruel, University of Zaragoza, 50009 Zaragoza, Spain; anolasco@unizar.es (A.N.H.); lgracia@unizar.es (L.G.S.)

**Keywords:** bullying, aggressor, victim, normality, diversity, role, violence, school life

## Abstract

Bullying is related to several variables, including diversity and variables that place the victim outside of normality. However, it is not easy to find a single meaning of normality. The present study has two main objectives: to find out whether victims are evaluated as non-normal and to find out whether aggressors are evaluated as non-normal. A cross-sectional, correlational, and quantitative study was designed, focusing on a representative sample of secondary school students from the Community of Madrid. The sample consisted of 2076 participants and was constructed using a stratified, proportional, and random sampling technique. To gather this information, a questionnaire was constructed. It includes a first section where sociodemographic and normality information is collected, and a second section made up of the Defensor del pueblo-UNICEF Bullying Questionnaire. The reliability and consistency of the questionnaire are acceptable (Cronbach’s alpha 0.91). For the comparison of means between groups, a Student’s *t*-test was applied, and the correlation between variables was calculated by applying the bivariate correlation test. Results show that victims are evaluated as non-normal while aggressors are perceived as normal. This implies that the risk of being involved in bullying situations as a victim can be predicted.

## 1. Introduction

Bullying was described as “a behaviour of harassment and physical, psychological or moral aggression carried out by one pupil, or a group of pupils, towards another, with an imbalance of power and in a repeated manner” [1]. In this first definition, most of the features that characterize a bullying situation already appear: it includes different types of violence, it is not an isolated event, but it is repeated over time, the aggressor has the intention to cause harm to their victim, and there is an imbalance of power between aggressors and victims that allows the aggression to take place [1] (p. 762).

As a result of the development of bullying research, several variables related to the possibility of participating as aggressors or as victims of such violence have been found [2]. Some expert authors point out that, regarding victims, although there are specific characteristics that appear to be linked to this role, these are not, in any case, necessary or sufficient reasons to explain the aggression [3]. As for the aggressors, when asked about the reasons why they assaulted their victims, they refer to the presence of certain variables related to physical appearances, such as the use of glasses or obesity, but also skin color, hair color, or the existence of a disability [4,5]. These variables are related to the concept of diversity and also to the concept of normality.

In a previous study, Carretero Bermejo et al. [4] asked whether victims of bullying perceived themselves as having any characteristic that placed them outside of normality, finding that indeed, people who perceive themselves as having some characteristic that places them outside the norm are involved more frequently and more intensively in bullying situations as victims than people who do not perceive themselves as having anything out of normality. In this way, victims partly explained their involvement in bullying situations in the victim’s role based on those characteristics assessed as not normal [6]. Based on these results, the question arises as to whether victims who perceive themselves as having a characteristic that places them outside the norm also perceive other victims as having some non-normal characteristic. Directly related to this question, we also ask ourselves whether the aggressors assess that the victims of their aggression in educational contexts have some trait that forms part of their definition of non-normality. If victims and aggressors coincide in pointing out the victims as non-normal, we would be in a situation where normality, or non-normality, would partly explain the involvement in bullying situations as a victim. Therefore, we would have valuable information for prevention and intervention in these situations.

In this study, we questioned the relationship between involvement in bullying as victims and the assessment of non-normality made about those victims by aggressors, as well as by people who are or have been victims, where the meaning of normal and non-normal is established by the context of reference, in this particular case, the school and the class group [4,7].

While it is true that we are all different, it is also true that not all differences have the same value nor are they included in the definition of normal, given that this definition goes beyond the merely statistical and in this way, very infrequent traits are conceived as positive, and therefore within the standard and desirable, and much more frequent traits are evaluated as negative, outside the normal, and undesirable [8]. For this study, we were mainly interested in the assessment that each person makes of normality or abnormality (intrapsychic or subjective model of normality) and not so much in the specific characteristics on which this assessment is based, given that we intended to find if there is a relationship between bullying and the assessment of normality or being out of the normal, and not the relationship between bullying and specific characteristics. Regarding this second point, several studies have been carried out in which the results show a relationship between specific physical, personality, or background characteristics and involvement in bullying situations [9,10].

If this is the case, on the one hand, we could be looking at victims of bullying who explain, and perhaps justify, the violence that they and the other victims receive based on characteristics that distance them from normality and aggressors who justify their aggressive behavior on certain characteristics that the victims themselves present [11], which are valued as not normal, and thus blaming them for the violence they receive [3]. It should be noted that it is not new for the dominant group to dictate and impose its definition of normal and try to control the rest of meanings, realities, or contents, through violence, if necessary, assigned to this normality. Why should it be any different in schools?

## 2. Theory

### 2.1. Bullying

Since bullying was first defined, many different definitions have been presented. However, in all these definitions, three recurring characteristics determine and explain the difference between bullying and other types of violence: bullying includes different types of violence and the intention to harm the victim; it is not an isolated behavior but appears recurrently, and it is possible due to an imbalance of power between victim and aggressor that the aggressor decides to abuse [3,12]. Similarly, progress in the study of bullying has facilitated the typification of different forms of violence and how it is carried out. From this perspective, and taking into account the way in which bullying is carried out, we can speak of: direct, in which there is a direct confrontation between both actors, as occurs in physical or verbal aggression; and indirect, in which aggression is not presented openly, but takes more subtle forms, such as false rumors, exclusion from groups of friends, or even *cyberbullying*, among other forms [13] (p. 619).

If we make a classification according to the type of violence [13,14], we find that bullying can be: physical, which includes pushing, punching, kicking, hitting, and burning; verbal, mainly involving name-calling and insults, although it is also common to belittle the victim in public, highlighting their physical defects or their actions; and psychological, which affects the victim emotionally, undermines their self-esteem, and generates a feeling of insecurity and fear. It involves intimidation and verbal and non-verbal threats. The psychological component can be found in any of the forms of mistreatment. There is also social dimension to bullying, where the aim is to isolate the victim, the aggressors try to eliminate the reputation that the victim may have in the group in order to achieve their social exclusion, and sometimes even try to make others participate in this type of harassment. This can be carried out through name-calling, mocking and spreading negative rumors, and sexual harassment, which involves sexual conduct that denigrates the victim.

The advancement of technology has also influenced the emergence of new forms of bullying [15], adding the following to the previous forms: dating violence and bullying between adolescent couples, where emotional blackmail prevails. It is considered the prelude to gender-based violence. Another new form of bullying is cyberbullying, an aggressive and intentional act carried out repeatedly and continuously over time, through the use of electronic forms of contact by a group or an individual against a victim who cannot easily defend themself [15,16].

There are three main roles in peer bullying situations: aggressor, victim, and bystander [15]. As a result of the in-depth study of peer bullying, different types or subtypes of these three main roles have been recognized and identified [17].

**Aggressor:** We can find three forms of aggressor: *leader aggressor*: students who initiate the aggression in peer bullying situations; *follower aggressor*: students who do not initiate the aggression, but who join the leader aggressor; *reinforcer aggressor*: students who participate in the situation by encouraging the aggressors and teasing the victim [18].

**Victim:** We differentiate between two possible types: passive victim and active victim [15].

**Bystander:** There are two basic types of bystander [14]: *defensive bystander*: students who help or try to help the victim; *passive bystander*: students who do not get involved in the situation.

### 2.2. Normality

If we look up the definition of “normal” in the Spanish Royal Academy of Language dictionary, we find the following definition: “that which serves as a norm or rule, which conforms to certain norms fixed in advance”. Searching for the meaning of “norm” is defined as the “rule or organisation of behaviour dictated by an authority, a rule that must be followed or to which behaviour, tasks or activities must conform”. If we look for “normalisation”, we find the definition of “action and effect of normalising” and, finally, if we look for the definition of “normality” we find that it is the “quality or condition of normal” [19].

It is not an easy task to determine a definition of normal, since it is conditioned by the authority that defines it and by the context in which it is reproduced. This implies that any change in authority or context can modify and even completely change the meaning of normal. For this reason, it is unlikely to find a shared definition of normality in different contexts, times or group [7].

The study of normality has been approached from different prisms, which has given rise to different models or perspectives of study of normality, depending on where each of these models puts the focus. First, from a statistical perspective, normality is defined by the midpoint, frequency, and continuity. The statistical model is presented as the most common, and the most used, within each cultural frame of reference [10]. From this model, the normal is identified with the most likely and frequent, the recurrent or with the highest probabilities of happening. The abnormality is then related to terms such as deviation, which represent everything that is not part of the normality defined from statistics. From this perspective, each group, in each context, defines its normality and, in addition, this normality is the only one possible and accepted by the group. Being part of this normality implies assuming, on one’s own, the characteristics that make up this definition of normality. However, this is not always possible. We can modify behaviors and attitudes, with the conflicts that these changes can generate but, nevertheless, it is not possible to modify most of our physical traits to be part of normality, to give an example.

The processes of discrimination against those who are different are part of the life of groups since the mere fact of being part of a group generates processes of exo-group discrimination [20]; thus, by identifying ourselves as members of the normal majority group, we tend to discriminate against all those who are outside this group defined as normal.

The second model, the legal model, pays attention to the normative, with the normal being that which conforms to the norms dictated in each context. Once again, there is a need to link normality with the social context of reference. From a third model, the medical model, the term normality is linked to health, where normality is considered an optimal state of health or conforms to what is expected based on different criteria used to assess normality (often statistical). Finally, from the subjective or intrapsychic model, it is proposed that it is each individual who defines and assesses their normality or abnormality [10].

Assuming these difficulties, we can ask ourselves what is considered normal and, consequently, what is accepted in our immediate context. According to some authors, if we refer to physical characteristics, we must also refer to those not usually made visible in the media, stories, or textbooks, characteristics such as weight, skin color, height, or the shape and hair color [19]. We all possess different characteristics that make us different, but only some of them [21], such as being Caucasian and slim, are considered legitimate and positive in our present occidental society. Meanwhile, some others, such as overweight, shortness, different skin color or eyes with strabismus, are not. Moreover, as we mentioned in the introduction, it is not merely a matter of statistics but also the dominant group’s decision and assumptions.

The concept of normality is a social construction, a representation of each group and its way of thinking, which is used in the same way in environments that have nothing to do with science, where it is used to facilitate the understanding of the social context and, also, in scientific and academic environments. This implies that the definition of normality is not typical of a single specific field or discipline. On the other hand, in relation to the above, the meaning of normality of is exclusive to the scientific field. In reality, all definitions of normality are intimately related to contents closer to common sense than to science. Therefore, being a widely used concept, it is not possible to construct a single definition [22].

## 3. Current Study

The objectives of this research are: firstly, to find out the perception of normality or non-normality that aggressors, victims and bystanders have of victims and aggressors; secondly, to find out whether there is a relationship between bullying and its different roles and the assessment of non-normality in victims and aggressors and the possible predictive role of the non-normality variable in participation in bullying situations as aggressor or victim. In relation to these objectives, the working hypotheses are:

**Hypothesis** **1** **(H1).**
*People who are involved in bullying as an aggressor perceive victims as non-normal and score significantly higher on victim abnormality than people who are not involved in bullying as an aggressor.*


**Hypothesis** **2** **(H2).***People who are involved in bullying as victims perceive victims as non-normal and score significantly higher on victim abnormality than people who are not involved in bullying as victims*.

**Hypothesis** **3** **(H3).**
*People who are involved in bullying as bystanders perceive victims as non-normal and score significantly higher on victim abnormality than people who are not involved in bullying as bystanders.*


**Hypothesis** **4** **(H4).***People who are involved in bullying as aggressors perceive aggressors as non-normal and score significantly higher on aggressor abnormality than people who are not involved in bullying as aggressors*.

**Hypothesis** **5** **(H5).**
*People who are involved in bullying as victims perceive bullies as non-normal and score significantly higher on aggressor abnormality than people who are not involved in bullying as victims;*


**Hypothesis** **6** **(H6).**
*People who are involved in bullying as bystanders perceive bullies as non-normal and score significantly higher on aggressor abnormality than people who are not involved in bullying as bystanders.*


## 4. Method

For the development of this study, a descriptive and correlational quantitative cross-sectional design has been implemented, in which a representative sample of students enrolled in Compulsory Secondary Education in the Community of Madrid has participated.

### 4.1. Sample

In the 2019–2020 academic year, the Community of Madrid had 281,430 students enrolled in Compulsory Secondary Education, distributed among public schools (*n* = 146,802), subsidized schools (*n* = 105,900) and private schools (*n* = 28,728). To ensure representativeness, the sample was selected using a stratified, proportional, and random sampling technique, taking into account the proportionality of the type of school and the distribution of students in the different districts of the Community of Madrid. A total of 2415 students participated, of which 2076 students completed the questionnaire correctly, this being the final sample in this design (Table 1). The participants’ ages were between 13 and 16 years old, with a mean age of 14.14 (standard deviation = 0.89). In terms of gender, 43% are girls, and 57% are boys (Table 1). The selected sample allows us to work with a confidence interval of 95% (significance of 0.05 is assumed) and a margin of error of 2.14.

This design studied the relationship between the independent variables, aggressor, victim, and bystander, and the dependent variable evaluation of victims and aggressors as normal or not. The variables age, grade, gender, type of school, and district were controlled.

### 4.2. Instrument

In order to carry out this research, a questionnaire with two different parts was developed. The first part collected socio-demographic information, the dependent variable, and the control variables of the research design. The second part of the questionnaire was intended to collect information on the independent variable, bullying. In the conduct of this study, to collect information on participation in situations of bullying, the Defensor del Pueblo-UNICEF [8] questionnaire was used. The constructed questionnaire had the following structure:

#### 4.2.1. Section 1: Socio-Demographic Variables

Dependent variable: Non-normality assessment of victims and aggressors. Two items were included in which participants were asked to evaluate between 0 and 10 the normality of the victims of bullying in the first of these items, and the normality of the aggressors in situations of bullying in the second, where 0 means completely normal and 10 completely non-normal.

Controlled variables:AgeGradeSexType of schoolDistrict

#### 4.2.2. Section 2: Defensor del Pueblo-UNICEF Questionnaire

It is a Likert-type scale with 39 items with choices ranging between 1 (never) and 4 (always). Items 1 to 13 measure involvement in bullying as a victim, items 14 to 26 measure involvement in bullying as an aggressor, and items 27 to 39 measure involvement in bullying as a bystander. For each of these roles, the questionnaire collects information on six dimensions of bullying: verbal aggression, indirect physical aggression, direct physical aggression, exclusion, sexual harassment, and threats.

## 5. Procedure

Once the schools and participants were selected, a meeting was held with each center’s management team to explain the project’s content and confirm their participation. Once their participation had been confirmed, the participating classes, the date, and the data collection schedule were randomly selected. An authorization form was provided so that each participant’s legal guardian could authorize their participation. Once in the classroom, the researcher explained the questionnaire’s content to each group, and 30 min were allocated to carry out the test.

The questionnaires that were not correctly completed have been eliminated from the total; the database has been created in the statistical programme SPSS v26 (IBM: Ciudad Real, Spain) and the data collected were entered. The sample was analyzed statistically and descriptively, the necessary variables for the study have been created, given that the questionnaire provided information on items that needed to be transformed into the variables under analysis. The questionnaire’s reliability and consistency were checked by calculating the Cronbach’s alpha for each of the sections and the test as a whole. To compare the results and the statistical significance of the differences found between the mean scores of the groups of the independent variable in the dependent variable, we used the Student’s *t*-test for independent samples. To study the possible relationship between the variables, we used the bivariate correlation test. To be specific, Pearson’s correlation coefficient was chosen. The Kolgomorov–Smirnov test was first applied to confirm whether the data of the sub-factors of each variable were the typical ones of a normal distribution. To assess whether or not students are involved as aggressors, victims, or bystanders in bullying situations, the sample score’s first quartile has been established as a mark [23]. To assess as a perception of non-normality, scores between 5 and 10 are assumed [10].

## 6. Results

Cronbach’s alpha was calculated to assess the reliability and consistency of the questionnaire used. The questionnaire shows adequate internal consistency and reliability (Cronbach’s alpha 0.91).

The results are presented grouped on the non-normality rating of victims first, followed by aggressors.

The results show statistically significant differences in the mean scores on non-normality of victims between aggressors and non-aggressors, victims, and non-victims, and bystanders and non-bystanders with higher scores in all cases for people involved in bullying situations. In the case of aggressors and victims, victims are rated as non-normal (Table 2).

Results show a high correlation between the variables aggressor and victim abnormality and victim and victim abnormality and a moderate correlation between bystanders and bystander abnormality.

According to the results found (Table 3), we can confirm the hypotheses H1: “People who are involved in bullying as aggressors perceive victims as non-normal and score significantly higher on victim abnormality than people who are not involved in bullying as aggressors” and H2: “People who are involved in bullying as victims perceive victims as non-normal and score significantly higher on victim abnormality than people who are not involved in bullying as victims”. We cannot confirm the hypothesis H3: “People who are involved in bullying as bystanders perceive victims as non-normal and score significantly higher on victim abnormality than people who are not involved in bullying as bystanders”.

Results indicate significant differences in the mean scores on the abnormality variable only in the case of the victim variable, where people who have been victims score higher than people who have not been victims (Table 4).

The results show a low correlation between the variables victim and bystander with aggressor abnormality. The correlation between aggressor and aggressor abnormality is very low.

According to these results (Table 5), we reject the hypotheses H4: “People who are involved in bullying as aggressors perceive aggressors as non-normal and score significantly higher on aggressor abnormality than people who are not involved in bullying as aggressors” and H6: “People who are involved in bullying as bystanders perceive aggressors as non-normal and score significantly higher on aggressor abnormality than people who are not involved in bullying as bystanders”. We cannot confirm our hypothesis H5: “People who are involved in bullying as victims perceive aggressors as non-normal and score significantly higher on aggressors abnormality than people who are not involved in bullying as victims”.

## 7. Discussion of the Results

According to the results found in this study, the aggressors perceive that the victims of bullying situations have some trait that defines them as non-normal, excluding them from normality. These results indicate the possibility that bullying finds justification in what is not normal, in what seems to be out of normality [19]. Historically, the dominant group, the “normal”, uses all the tools at its disposal, including violence, to maintain and impose this normality vision. The school context could be reproducing what happens in the social context in which it is immersed [17]. In this research, victims also rated victims as non-normal. If aggressors could justify their aggressions based on the non-normality with which they evaluate the victims [24], these victims could justify the violence they receive based on the same non-normality they report themselves in [2].

Different studies [5,14] point out that any element that singles out the pupil and differentiates him/her from the group, in general, can be a reason to be ridiculed and victimized by the aggressor. It is the subject of further work to find the possible variables related to the victim’s position. However, it seems that it is not so much the characteristic itself that matters as assessing that victim and aggressor’s characteristic since the definition of normal is shared [25]. This assessment of normality is not only subject to statistical criteria [4,7], which seems to be behind the aggressions [26]. From the results found in this study, it seems that both aggressors and victims justify the aggression based on the normality of the victims. We consider this information relevant for elaborating projects for the intervention and, especially, prevention of bullying.

This study, on the other hand, presents the aggressors as normal, with no traits evaluated as non-normal by any of the current roles in bullying situations, not even by the victims, which reinforces the hypothesis that the normalized group, the normal, is the one that defines normality and exercises the necessary actions to maintain it and punish those who do not conform to that normality [5]. These data could explain why the victims themselves evaluate the victims as non-normal and the aggressors as normal, even though they are assaulting them.

Bystanders do not find abnormal traits in aggressors or victims, which may be behind the supportive or neutral behavior that bystanders tend to adopt when facing the aggressions they witness. What bystanders think and do is crucial as they are the most frequent role and the one in which most students are involved [16]. The role of the bystander depends, in part, on whether the aggressors find the necessary support to develop their aggressive behavior or whether the victims find allies to confront the violence they suffer and to prevent these situations from occurring [14]. In this sense, working with the group of bystanders, with the group that surrounds the victims, is essential when implementing intervention and prevention programs against bullying situations.

It was not the aim of this research to study possible differences between the different types of aggressions. However, we found that all types of aggressors and all types of victims presented very similar results [26]. Furthermore, this research does not address the study of which traits are most frequently found in the victims nor explain the aggressions, since, as we have explained, it is not so much the trait that matters as the evaluation of normal or non-normality that is made of such characteristics.

### 7.1. Limitations of the Research

This design describes a very specific situation: aggressors and victims evaluate the victims as not normal and the aggressors as normal. However, they were not asked whether this non-normality explains the violence they exert or the aggressions they suffer so that this possible relationship is a hypothesis to be studied and not a confirmed relationship.

This study was developed with specific self-report tests, which generated at least two limitations: the first related to the influence of social desirability during the process of answering the questionnaires; the second related to the influence that the usage of one test or another may have on the results, even though they both aim to measure the same thing, in this case, bullying.

Our sample represents Compulsory Secondary Education (ESO) students in the Community of Madrid; generalizing these results to the population of ESO students in the rest of the country or other countries was not possible with this single study.

This study addresses the relationship between normality and participation in situations of bullying, but has not taken into account the different types of bullying that can occur and whether this relationship is modified by the specific type of harassment in question.

Finally, 339 questionnaires were eliminated because they were not correctly performed, and it was not possible to establish the influence of these lost results on the final results of this study.

### 7.2. Future Lines of Work

Concerning the research, four lines of work were proposed.

Firstly, and as identified in the limitations, we propose to replicate this study in the different communities to contrast the results found. It may be necessary and interesting to replicate this study also in different countries. We also consider it necessary to evaluate the relationship found in this work according to the type of bullying that is executed or receive. In this sense, and given the increasing frequency of cyberbullying situations, it does not seem necessary to include this type of violence in future studies. Secondly, it is proposed to explore several questions: Do victims and aggressors justify violence in terms of their characteristics in terms of not being within what is perceived and valued as normal? If so, how do they elaborate their justifications? What variables define normality in our school context? Which of these are related to involvement in bullying situations as a victim?

Thirdly, the importance of including the teaching staff in future designs is considered, as they are key actors in identifying and intervening in bullying situations.

In terms of intervention, we believe it is essential to include the information gathered in this and other studies about the variables behind bullying situations, such as gender and sexism. This means that it is not enough to work on certain characteristics, but it is necessary to include content related to acceptance, integration, diversity, and normality in prevention and intervention programs.

## 8. Conclusions

The fact that aggressors and victims define victims as non-normal and aggressors as normal in the same way would seem to indicate that there is a group that determines what is and what is not normal and has the capacity, with the possible support of the bystanders, to maintain and impose it, in such a way that this normality, or non-normality, serves to justify the violence against the victims. Moreover, these aggressions seem to be justified by the aggressors and also by the victims, given that, we insist, the victims also evaluate themselves as non-normal and evaluate the aggressors as normal. It is worth questioning whether, besides agreeing on the definition of who is and who is not normal, they also agree on the definition of normal and, above all, whether this non-normality explains the aggression perpetrated by the aggressors and received by the victims, as the results of this study seem to suggest.

We cannot ignore the fact that bullying situations are, to a large extent, made possible by the direct and indirect support and reinforcement that the aggressor finds in their reference group, in the form of aggressor profiles without the capacity to lead and start the aggression or in the form of passive bystanders [14]. Furthermore, these bystanders consider the aggressors to be normal or with no abnormal traits. This is why bullying intervention and prevention processes consider and must consider the bystander’s role as a fundamental agent of change and protection, given that without the involvement of this majority, many forms of bullying would not be possible.

We consider that, according to the results found, the challenge of building an inclusive and diverse school system, where differences are not evaluated negatively, would also have a positive impact on the reduction of bullying situations if, as the results seem to indicate, these variables associated with what is different, with what is normal, are related to involvement in bullying situations as a victim.

Finally, these results reflect a change of view from the victim to the aggressor: drawing up lists of characteristics presented by victims of bullying can generate processes of revictimization, including justification of the violence that victims receive based on their personal characteristics. Now, instead, we can also work with the aggressors and their support group, with their beliefs about normalcy and diversity.

## Figures and Tables

**Table 1 ijerph-19-06344-t001:** Description of the sample.

	Sex		School			Grade			
	Boys	Girls	Public	Subsidized	Private	1	2	3	4
N	1183	893	981	715	380	451	589	502	534
Percentage	57%	43%	47.25%	34.40%	18.35%	21.7%	28.3%	24.1%	25.9%

**Table 2 ijerph-19-06344-t002:** Perception of abnormality in victims of aggressors, victims, and bystanders.

		Aggressor		Victim		Bystander	
		Yes	No	Yes	No	Yes	No
**Abnormality victims**	N	791	1519	1425	885	1640	670
	Mean	7.12	2.47	5.95	2.70	4.33	2.24
	Sig. Bilateral	0.000		0.000		0.000	

**Table 3 ijerph-19-06344-t003:** Linear correlation of aggressor, victim, and bystander with victim abnormality.

		Aggressor	Victim	Bystander
**Abnormality victims**	Pearson Correlation	0.795	0.701	0.460
	Sig. Bilateral	0.000	0.000	0.000

**Table 4 ijerph-19-06344-t004:** Perception of abnormality in aggressors of aggressors, victims, and bystanders.

		Aggressor		Victim		Bystander	
		Yes	No	Yes	No	Yes	No
**Abnormality aggressors**	N	791	1519	1425	885	1640	670
	Mean	2.12	2.17	3.43	3.04	2.36	2.45
	Sig. Bilateral	0.067		0.034		0.132	

**Table 5 ijerph-19-06344-t005:** Correlation of aggressor, victim, and bystander with aggressor abnormality.

		Aggressor	Victim	Bystander
**Abnormality aggressors**	Pearson Correlation	0.097	0.253	0.174
	Sig. Bilateral	0.000	0.000	0.000

## Data Availability

The database is available to those researchers who intend to replicate the investigation and need its consultation, upon submission of the application and justification of that need to the author of correspondence. The SPSS v.26 application was used for data analysis in this investigation.

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
