# Peer review of "Non-Normality as a Predictor of Participation in Bullying: Valuation in Victims and Aggressors"

_ijerph, 2022, doi:10.3390/ijerph19106344_

Round 1

Reviewer 1 Report

Introduction

  • The authors begin the paper by stating “One of the first definitions of the term bullying was described as …..” and provides a citation for a 2019 article. As it is written it is a little confusing since it is clear that the definition was not originally posed in that article. The sentence should be re-written to cite the original citations where the first definition was provided or the authors should remove the part that states “first definition.”
  • The references should be numbered in order of appearance in the text. Yet, the first citation is listed as [18], followed by [11], [10], [2] and [22]. The numbering of the citations continues to be off as the paper progresses. The authors need to re-do the in-text citation numbering and the full reference list.
  • On page 2, line 45, the authors note “In a previous study, we asked whether victims of bullying …” This previous study needs to be cited. Also, the authors should avoid stating “we asked” and instead note the citation for the actual study. For example “In a previous study, Author et al [xx] asked whether victims ….”
  • The study has multiple hypotheses and they are currently presented as one very long sentence. The authors should consider re-writing the hypotheses of the study to separate them which would make it easier for the reader to visualize and for the authors to make reference to in the results and discussion section. For example:
    • H1: People who are involved in bullying as an aggressor perceive victims as non-normal and score significantly higher on victim abnormality than people who are not involved in bullying as an aggressor.

      H2: People who are involved in bullying as victims perceive victims as non-normal and score significantly higher on victim abnormality than people who are not involved in bullying as victims.

      And so forth numbering the remaining hypotheses in order (H3, H4, etc.)

  • Section 2 (2.1 Bullying and 2.2 Normality) should be presented before the study hypotheses are introduced. The authors can consider moving the paragraphs where the objectives of the study are discussed to be a standalone section titled “The Current Study” and present that at the end of the literature review before the “Methods” section.
  • In line 196, the authors use the term “fatness,” the authors should consider changing this to “overweight” or “obesity” since the term fatness has a judgmental tone whereas the suggested terms are more neutral and presented in a medical context.
  • The authors state that “We all possess different characteristics that make us different, but only some of them [2], such as being Caucasian and slim, are legitimate and considered positive in our present society.” This is not true in all contexts – being Caucasian and slim may be considered positive in a Western context or specific contexts. Some parts of the world may not see this as positive. Please consider re-framing the sentence or temper the statement to clarify which “present society” they are referring to. 

Method

  • The authors used a stratified, proportional and random sampling technique. If there are any, additional detail should be provided to explain the inclusion and exclusion criteria for study participants.
  • In the sample section, the authors state that 2415 students participated, and 2076 completed the questionnaire correctly. Thinking of the original pool of students that had the option of being recruited, were there any that refused to participate or did not participate for a specific reason? A brief consideration of a non-response bias and its potential on the results of the study should be included, at a minimum in the limitation section of the paper.
  • The authors need to present data about the socio-demographic variables (age, grade, sex, type of school, district) in a table.
  • It would be too much to include in the body of the manuscript, however, the authors should consider listing the 39 items from the Defensor del pueblo-UNICEF Questionnaire as part of an appendix (if the journal allows it). If not, then at a minimum, the authors should describe some of the key items in the text so that the reader understands what kinds of questions were used. Of particular importance is the need to describe the key dependent variable “Non-normality assessment of victims and aggressors.” It is unclear what this entailed and that is the main focus of the study. This needs to be included in the main text.

Results

  • The hypotheses should be numbered at the beginning of the paper and in the results section the authors can refer to each hypotheses and whether it was supported or rejected. For example, “According to the results found (Table 2), we can confirm that H1: People who are involved in bullying as aggressors perceive victims as non-normal and score significantly higher on victim abnormality than people who are not involved in bullying as aggressors and H2: People who are involved in bullying as victims perceive victims as non-normal and score significantly higher on victim abnormality than people who are not involved in bullying as victims are both supported.”
  • The questionnaire collected information on six dimensions of bullying - verbal aggression, indirect physical aggression, direct physical aggression, exclusion, sexual harassment and threats. Did the authors analyze the differences across the various forms of bullying? I think it would be important to know whether victims and aggressors’ perception of abnormality varies based on the type of bullying behavior. This could be an additional analysis that is included in the study.

Limitations

  • The authors correctly note that “generalising these results to the population of ESO students in the rest of the country is not possible with this single study.” Given the international readership of the journal and the relevance of the study, the authors should change this to state “generalizing these results to the population of ESO students in other parts of Spain or other countries is not possible with this single study.”

Future lines of work

  • As noted above, the authors should examine how the perception of abnormality differs based on the type of bullying. However, it seems the current questionnaire did not collect information on cyberbullying. Future studies could also examine this line of inquiry.

Overall, the authors provide an important and novel contribution to the bullying literature. It has key policy implications especially as it relates to bullying prevention and interventions. I hope these comments are constructive and useful to help improve the paper. 

Reviewer 2 Report

First of all, I thank the editor for giving me the opportunity to review this paper.

The topic is very topical and extremely important, despite the difficulties that are always encountered in describing normality and non-normality. Moreover, this type of work represents a unique opportunity to learn about local realities and compare them in an international context in future research. 

The methodology is well presented, well used and quite simple but adequate to the data. However, the use of self-report instruments may have interfered with the results given the high social desirability in these situations. 

The relationship with the different explanatory concepts that the authors insert in the discussion, which are often disconnected and not explained in previous sections such as the introduction, results, etc., needs to be improved. 

Reference could be made to international comparative data. 

Little depth is given to future lines, as well as to the usefulness of the results in violence prevention processes. It is only mentioned in one line. 
